# Targeted Disruption of the MORG1 Gene in Mice Causes Embryonic Resorption in Early Phase of Development

**DOI:** 10.3390/biom13071037

**Published:** 2023-06-24

**Authors:** Sophie Wulf, Luisa Mizko, Karl-Heinz Herrmann, Marta Sánchez-Carbonell, Anja Urbach, Cornelius Lemke, Alexander Berndt, Ivonne Loeffler, Gunter Wolf

**Affiliations:** 1Department of Internal Medicine III, Jena University Hospital, 07747 Jena, Germanygunter.wolf@med.uni-jena.de (G.W.); 2Medical Physics Group, Institute of Diagnostic and Interventional Radiology, Jena University Hospital, 07747 Jena, Germany; karl-heinz.herrmann@med.uni-jena.de; 3Department of Neurology, Jena University Hospital, 07747 Jena, Germanyanja.urbach@med.uni-jena.de (A.U.); 4Institute for Anatomy I, Jena University Hospital, 07743 Jena, Germany; corneliuslemke@gmx.de; 5Institute of Forensic Medicine, Section Pathology, Jena University Hospital, 07743 Jena, Germany; alexander.berndt@med.uni-jena.de

**Keywords:** MAP kinase organizer protein 1, WDR83, WD-40 repeat domain 83, PHD3, ERK

## Abstract

The mitogen-activated protein kinase organizer 1 (MORG1) is a scaffold molecule for the ERK signaling pathway, but also binds to prolyl-hydroxylase 3 and modulates HIFα expression. To obtain further insight into the role of MORG1, knockout-mice were generated by homologous recombination. While Morg1+/− mice developed normally without any apparent phenotype, there were no live-born Morg1−/− knockout offspring, indicating embryonic lethality. The intrauterine death of Morg1−/− embryos is caused by a severe failure to develop brain and other neuronal structures such as the spinal cord and a failure of chorioallantoic fusion. On E8.5, Morg1−/− embryos showed severe underdevelopment and proliferative arrest as indicated by absence of Ki67 expression, impaired placental vascularization and altered phenotype of trophoblast giant cells. On E9.5, the malformed Morg1−/− embryos showed defective turning into the final fetal position and widespread apoptosis in many structures. In the subsequent days, apoptosis and decomposition of embryonic tissue progressed, accompanied by a massive infiltration of inflammatory cells. Developmental aberrancies were accompanied by altered expression of HIF-1/2α and VEGF-A and caspase-3 activation in embryos and extraembryonic tissues. In conclusion, the results suggest a multifactorial process that causes embryonic death in homozygous Morg1 mutant mice, described here, to the best of our knowledge, for the first time.

## 1. Introduction

Mitogen-activated protein kinase organizer 1 (MORG1), also known as WDR83, is composed of 315 amino acids and belongs to the WD-40 domain protein family, which function as adapters in different protein–protein or protein–DNA complexes [1]. The MORG1 gene was shown to be expressed in various tissues, including the heart, brain, kidney, testis and liver [2]. Over the past few years, MORG1 was identified in a wide variety of physiological signaling pathways as well as in diverse pathophysiological processes.

MORG1 was also characterized as a binding partner of the MP1 scaffold protein that facilitates the activation of MEK1 and extracellular signal-regulated kinase 1 (ERK1). MP1 binds to late endosomes via the adapter protein p14, regulating endosomal traffic, cell proliferation and tissue homeostasis during embryogenesis [3]. MORG1 also interacts directly with different components of the mitogen-activated protein kinase (MAPK) cascade and stabilizes their assembly into an oligomeric complex. MORG1 selectively promotes the activation of ERK1/2 or modulates their activation depending on its intracellular concentration in response to various agonists [2].

In addition to its role in the MAPK pathway, MORG1 is involved in another major pathway, the hypoxia-inducible factor (HIF) signaling, that regulates adaptive responses under tissue hypoxia. Under hypoxic conditions, the labile alpha and stable beta subunit of the HIF molecule heterodimerize and form a functional complex. The complex then translocates into the nucleus, where it induces the transcription of various genes involved in angiogenesis, cell proliferation, apoptosis and glucose metabolism [4]. HIF-1α causes upregulation of glycolytic genes, which are responsible for tissue metabolic adaptation to oxygen deprivation and anaerobic ATP synthesis [5], whereas HIF-2α is critical for the improvement of oxygen supply to hypoxic regions/areas by inducing erythropoietin (EPO) and vascular endothelial growth factor (VEGF) [5,6]. Under normoxia, prolyl hydroxylases 1-3 (PHD1-3) catalyze the oxygen-dependent hydroxylation of HIFα subunits, triggering its proteasomal degradation. MORG1 is a stabilizer of PHD3 and, thus. causes enhanced degradation of HIF [7]. It was previously shown that suppression of MORG1 induces HIF-mediated reporter gene activity and increases the stability of HIF-1/2α proteins [7].

Furthermore, MORG1 is involved in the formation of apicobasal polarity in epithelial cells, a key step in tissue development and organization [8,9]. Finally, in humans, MORG1 was identified as an associated protein in the spliceosome, which catalyzes the pre-mRNA splicing [10].

While examining renal ischemia/reperfusion injury in mice, we previously observed that heterozygous Morg1+/− mice exhibited a stronger increase in HIF-1/2α expression in the kidney, associated with higher serum EPO levels. Interestingly, these animals were partially protected and showed less renal inflammation in response to acute renal ischemia [11].

Further studies also pointed to a partial protection of Morg1+/− kidneys against systemic hypoxia and renal injury as a late consequence of diabetes mellitus types 1 and 2 [6,12]. This renoprotection may result from MORG1-dependent alterations in lipid metabolism in the course of diabetic nephropathy [13].

All mentioned pre-clinical findings were obtained using heterozygous Morg1+/− knockout mice, since no homozygous (−/−) knockout animals are born. Indeed, the homozygous Morg1−/− knockout genotype is not yet characterized in detail; therefore, we reasoned that it could provide important clues as to the physiological function of MORG1 in tissue development and homeostasis. Here, we present a detailed study of Morg1−/− embryos. We report, to the best of our knowledge, for the first time on the exact timing of embryonic lethality of Morg1−/− knockout progeny, the characterization of initial malformations and altered signal transduction pathways in early embryonic development that are associated with early embryonic death. In addition, we provide information for using MRI investigations as an innovative tool to screen for embryonic abnormalities.

## 2. Materials and Methods

### 2.1. Construction of the Targeting Vector

A murine 129/Sv BAC genomic library was screened using murine Morg1 specific primers to generate an 810 bp amplicon. Two clones were isolated, restriction mapped and partially sequenced. An 8.4 kb restriction endonuclease BamHI fragment was subcloned into a cloning vector. This fragment spanned the whole coding region of the Morg1 gene. A positive selectable cassette was introduced at two NsiI sites to delete the first five exons and create the replacement vector pMorg1-KO. This cassette contained stop codons in all three frames, an internal ribosomal entry site (IRES), followed by the enhanced green fluorescent protein (EGFP) gene with a SV40 polyadenylation signal and a neomycin phosphotransferase gene (NEOr) with polyadenylation signal, under control of the mouse phosphoglycerate kinase (PGK) promoter. A negative selection cassette containing the diphtheria toxin (DTA) was ligated to the 5′ end of the construct.

Transfection of embryonic stem cells (ES), screening for homologous recombination and generation of Morg1 knockout mice.

To obtain Morg1 chimeric mice, the construct was linearized, then electroporated into ES and selected for resistance to geneticin (G418). The resistant ES clones were analyzed for homologous recombination at the 5′ and 3′ end by polymerase chain reaction (PCR). Positive identified clones were further screened for homologous recombination with the targeted vector.

The clones were injected into C57BL/6J blastocysts to obtain chimeric mice. Male chimeras were mated to C57BL/6J mice and offspring screened for transmission of the disrupted allele. The heterozygous mice were backcrossed for 8–10 generations into C57BL/6J mice. Morg1+/− were intercrossed to characterize phenotypes of Morg1 homozygous (−/−) knockout mice.

All animal experiments were approved by the Local Ethics Committee of Landesamt für Gesundheit und Verbraucherschutz Hamburg and were carried out in accordance with the German Animal Protection Law. The animals were housed in a pathogen-free facility with a 12 h light-dark cycle and raised on standard chow and water ad libitum. For all animal experiments, the ARRIVE guidelines 2.0 were followed.

### 2.2. MRI

For MRI examination, 4–5 implantation sites remained connected by the uterine horn. They were fixed in 4% paraformaldehyde overnight (o/n) and embedded in 1% low temperature melting agarose in a falcon to keep in place. The medium must not interfere with the imaging process, must be stable to heat and easy to remove after MRI. The position of the embryos in the falcon had to be as close to the CryoProbe coil as possible. The embedded embryos were stored at 4 °C before and after the examination.

The embryo samples were scanned on a 9.4T Bruker MRI system (Bruker GmbH, Ettlingen, Germany) using the 2 ch quadrature CryoProbe to optimize the signal. For imaging, two different T2 weighted spinecho sequences were used. First, a standard Bruker MSME sequence in 3D mode to achieve an isotropic resolution of 50 µm, acquiring 8 equidistant echoes starting at TE = 7.46 ms (deltaTE = 7.46 ms), repetition time was TR = 1200 ms and the total acquisition time TA = 8 h.

The second sequence was developed in house [14] using a RARE acquisition with variable Flipangles (varFlipRARE). The flipangles followed an introductory series to achieve a steady state at alpha = 30°, which was then slowly increased during the RARE echo train. This allowed a RARE factor of 61 for a highly efficient data acquisition, permitting an isotropic resolution of 37 µm at nominal TE = 78 ms and TR = 1500 ms, acquiring 12 averages within a total acquisition time of TA = 6 h 6 min. For these scan parameters, the varFlipRARE sequence showed a pronounced diffusion weighting, mostly suppressing the MRI signal from liquid water. After the MRI examination, the embryos were cleared from agarose, embedded in paraffin and sectioned as described in Section 2.4.

### 2.3. Genotyping

The material for DNA isolation was obtained in three different ways: from tail biopsies, Reichert’s membrane or paraffin sections. For genotyping of adult mice, tail biopsies were used. If the embryos underwent preparation, a piece of the Reichert’s membrane was removed and used for the isolation. The tail biopsies were digested o/n and the Reichert’s membrane samples for 2 h in proteinase A at 56 °C and gentle shaking in the Thermomixer C (Eppendorf, Hamburg, Germany). After the first incubation, the temperature was raised to 70 °C for 30 min and DNA was isolated using the NucleoSpin Tissue kit (Macherey-Nagel, Düren, Germany).

The embryos, which were embedded in utero, were sectioned first. A piece of embryonic material was then scratched off the slide using a pipette tip and diluted in xylene. The digestion and isolation were performed using the QIAamp DNA FFPE Tissue kit (QIAGEN, Venlo, The Netherlands).

Genotyping was performed by PCR using the Mastercycler gradient (Eppendorf, Hamburg, Germay) to detect the mutated (sense primer 5′-GGCAAGGGCCAGTCAGCCTGC-3′; antisense primer 5′-GCCTCTGTTCCACATACACTTCAT-3′) and the wild-type alleles (sense primer 5′-GGCAAGGGCCAGTCAGCCTGC-3′; antisense primer 5′-GATAACGAGGCAACACTTCATCCT-3′). The mastermix was composed of 10 µL GoTaq DNA Polymerase (Promega, Fitchburg, MA, USA), 1 µL of three different Morg1 primers and 4 µL or 7 µL of sample and nuclease-free water to a total of 20 µL. When DNA was isolated from paraffin sections, 7 µL of isolated DNA were used due to small tissue volume and subsequent low DNA amount in the sample. If the isolation was performed from tail biopsies or Reichert’s membrane, only 4 µL of isolated DNA were used and 3 µL of nuclease-free water added. Afterwards, the gene segments were analyzed by gel electrophoresis in the PowerPac Basic (Bio-Rad Laboratories, Hercules, CA, USA). In the evaluation, the wild-type band appeared at 500 bp, the knockout band at 300 bp and in the heterozygote samples, both bands can be observed. The gels were examined at the G-Box F3 (Syngene, Bengalore, India).

### 2.4. Preparation, Fixation, Embedding, Sectioning

The whole preparation was performed in a Petri dish containing PBS on ice. The embryos were separated from the surrounding extraembryonic tissue, placed in individual Petri dishes and rinsed in PBS. The uterus and other extraembryonic tissue were then removed under a stereomicroscope using pointed tip tweezers. A piece of the Reichert’s membrane was extracted for genotyping and stored at −20 °C until further processing. The embryos were then photographed and, subsequently, fixed according to different protocols.

The fixation was performed using 4% paraformaldehyde (PFA) solution (RotiHistofix; Carl Roth, Karlsruhe, Germany). The various organs of adult mice as well as the embryos, that were still in utero, were placed directly in a tissue cassette and fixed o/n. The extracted embryos were placed in the cap of an Eppendorf tube, which was wrapped in a tissue bag, and then placed in a tissue cassette. In that way, it was possible to prevent the embryos from being crushed directly in the bag or lost through the gaps in the cassette. For those embryos, the following PFA incubation times were applied: E8.5 = 1.5 h; E9.5 = 2 h.

After fixation, the organs and embryos were rinsed in tap water for 1 h. The paraffin wax embedding was performed using the automatic Tissue Processor Leica TP1020 (Leica Biosystems, Wetzlar, Germany). After the paraffinization, the samples were embedded in paraffin blocks (Wetzlar, Germany) and stored at 4 °C until further sectioning.

The 4 µm thick tissue sections were prepared using the Automated Rotary Microtome (Leica Biosystems, Wetzlar, Germany) and dried o/n at 37 °C.

### 2.5. Immunohistochemistry, Immunofluorescence

In order to prepare the sections for further staining, deparaffinization and hydrogenation had to take place. For heat-induced epitope retrieval, the slides were placed in preheated citrate buffer, consisting of 8.2 mL sodium citrate, 1.8 mL citric acid and 90 mL dH_2_O, and cooked in a steamer for 25 min. The slides were then briefly cooled in dH_2_O and washed in PBS. Deactivation of endogenous peroxidases was carried out for 10 min in 3% hydrogen peroxide (H_2_O_2_) solution (Carl Roth, Karlsruhe, Germany) at room temperature (RT). The sections were then washed three times for 5 min in PBS while shaking.

The blocking of nonspecific binding sites was achieved by incubation in RotiBlock (Carl Roth, Karlsruhe, Germany) blocking solution for at least 1 h at RT. After removal of excess blocking solution, primary antibody incubation took place in a wet chamber o/n at 4 °C. The following primary antibodies were used: rabbit polyclonal anti-CD31 (epitope C-terminus aa 650; Abcam, Camebridge, UK), rabbit monoclonal anti-Ki67 (epitope C-terminus; Abcam, Camebridge, UK), rabbit polyclonal anti Morg1 (Biotrend, Berlin, Germany), rabbit polyclonal anti-PHD3 (Invitrogen, Carlsbad, CA, USA), rabbit polyclonal anti-VEGF-A (Lifespan Biosciences, Seattle, WA, USA), goat polyclonal anti-HIF-1α (epitope Arg575-Asn826; R&D Systems, Minneapolis, MN, USA) and goat polyclonal anti-HIF-2α (epitope Ser542-Thr87; R&D Systems, Minneapolis, MN, USA). The following day, the slides were washed three times for 5 min in PBS while being shaken. The horseradish peroxidase (HRP) labeled rabbit anti-goat IgG and goat anti-rabbit IgG secondary antibodies (both Medac Diagnostics, Wedel, Germany) were diluted 1:500 in blocking solution and incubation took place in a wet chamber for 1 h at RT. After that, the slides were washed again in PBS in the same way as before. The ImmPACT NovaRED Substrate Kit (Vector Laboratories, Burlingham, CA, USA) was prepared according to the instructions. Incubation times varied between 3 and 10 min, depending on the intensity of the staining reaction. The slides were then washed again in PBS three times and the sections coverslipped using AquaTex (Merck Millipore, Burlington, MA, USA).

To ascertain the specificity of MORG1 antibody, different strategies were performed: (1) “negative tissue” control: staining of MORG1 knockout embryos; (2) “secondary antibody only” control: staining with buffer instead of primary antibody; (3) “isotype” control: staining with IgG control instead of primary antibody (see Appendix A).

For immunofluorescence, after deparaffinization, heat-induced epitope retrieval was performed by cooking the slides in preheated sodium citrate/citric acid buffer in a steamer for 15 min. The blocking of nonspecific binding sites was carried out for 1 h in blocking solution consisting of 5% BSA in PBS sterile. Primary rabbit anti-cleaved caspase-3 antibody (Cell Signaling, Leiden, The Netherlands) was diluted in blocking solution and incubated at 4 °C o/n in a wet chamber. The following day, the sections were washed three times for 10 min in PBS while being gently shaken and, subsequently, incubated in blocking solution for 30 min. The secondary goat anti-rabbit IgG antibody, labeled with DyLight594 (Vector Laboratories, Burlingham, CA, USA), was diluted 1:200 in blocking solution and incubated for 2 h at RT in a wet chamber. The sections were then washed in PBS three times for 10 min and incubated with DAPI (Sigma Aldrich, St. Louis, MI, USA) for 10 min at RT in a dilution of 1:2000 in PBS sterile. The final washing steps were caried out three times for ten minutes in a sterile filtrated solution of PBS and 0.1% TritonX-100 (Ferak Berlin, Berlin, Germany). Finally, the sections were coverslipped using Vectashield antifade mounting media (Vector Laboratories, Burlingham, CA, USA) and immediately examined.

### 2.6. Hematoxylin and Eosin Staining (H&E)

First, deparaffination in xylol for 15 min and rehydration took place, after which hematoxylin (Hemalaun Solution Acid acc. to Mayer; Carl Roth, Karlsruhe, Germany) staining for 5 min ensued. Following washing in tap water for 15 min, the slides were counterstained with eosin G (Carl Roth, Karlsruhe, Germany) for 5 min. The slides were covered using the xylene soluble mounting media Pertex (Medite, Burgdorf, Germany) and the Automatic Cover Slipping Machine (Medite, Burgdorf, Germany).

### 2.7. Microscopy

All sections from IHC, IF and H&E staining were examined at the ZEISS Axio Imager 2 (Carl Zeiss, Oberkochen, Germany) or with whole slide scanner Vectra^®^ Polaris™ (Akoya Bioscience, Menlo Park, CA, USA) to show whole embryos. The caspase-3 IF microscopy was performed using channels specific for AF594 and DAPI.

### 2.8. Multiplex Immunofluorescence Labeling (mIF) and Multispectral Imaging

For multiplex IF, 3 µm thick 12.5-day old wild-type murine embryo FFPE sections were placed on IHC microscope slides and dried for 1 h at 65 °C before deparaffination. Then, deparaffination and fixation in 4% PFA solution (RotiHistofix; Carl Roth, Karlsruhe, Germany) took place. For mIF labeling, the Opal™ 4-Color anti-Rabbit Manual IHC Kit plus Opal™ 620 (Akoya Bioscience, Menlo Park, CA, USA) was used according to the manufacturer’s protocol. After verification of the optimal pH of the AR buffer during HIER method as well as the optimal dilution for each primary antibody in a monoplex IF staining, the optimal staining sequence was tested by trying different positions of the primary antibodies. For multiplexing, the mandatory steps (HIER, blocking, incubation of the primary and secondary antibody, OPAL signal generation) were repeated for each primary antibody (rabbit polyclonal anti-Morg1 (Boster Biological Technology, Pleasanton, CA, USA); rabbit monoclonal anti-WT1 (epitope human WT1 aa 59–269; Invitrogen, Waltham, MA, USA); rabbit monoclonal Ki67 (Abcam, Cambridge, UK); rabbit monoclonal anti-CD31 (epitope total CD31; Cell Signaling Technology, Danvers, MA, USA)).

After a final heating cycle in AR buffer and staining with DAPI, the slides were mounted with ProLong™ Diamond Antifade Mountant. For imaging, the scanner Vectra^®^ Polaris™ (Akoya Bioscience) was used, which created whole-slide scans. Afterwards, the software Phenochart^TM^ was necessary to visualize the scan file and adapt the brightness and color of specific Opals™. This method allowed for simultaneous detection of distinct biomarkers in various tissues leading to spatial phenotyping of the sample.

The autofluorescence controls were performed by monoplex staining without primary antibody (see Appendix A).

Furthermore, using the simulated brightfield mode (Pathology Views^TM^), it was possible to render single channels of the fluorescence images displaying them as simulated DAB/Hematoxylin images.

### 2.9. Whole Mount Staining

Following the preparation, the embryos were rinsed in ice cold PBS and fixed in a 4:1 methanol/dimethyl sulfoxide (DMSO) solution o/n at 4 °C. The following day, the embryos were incubated for 5 h in a 4:1:2 methanol/DMSO/30% H_2_O_2_ solution. The H_2_O_2_ was used again to block endogenous peroxidase activity. After that, the tissue could be dehydrated and stored in 100% methanol at −20 °C or the staining could continue.

In order to proceed with IHC staining, the embryos must first be rehydrated. This was carried out in descending alcohol concentrations of 75%, 50% and 25% for 5 min in each solution while gently rocking. The embryos were then incubated twice in 0.5% TritonX 100 (Ferak Berlin, Berlin, Germany) in PBS sterile for 30 min at RT to further permeabilize the tissue. Permeabilization of the embryos was necessary for the blocking solution and antibodies to penetrate the tissue sufficiently. The blocking of nonspecific binding sites was achieved by incubation in RotiBlock blocking solution containing 0.5% TritonX 100 (Ferak Berlin, Berlin, Germany) and 0.2% sodium azide (Carl Roth, Karlsruhe, Germany) at 4 °C o/n. The embryos were then washed twice for 10 min in blocking solution. Embryos were placed in a 2 mL Eppendorf tube with rabbit anti-MORG1 antibody (Biotrend, Köln, Germany) diluted 1:1000 in blocking solution containing 0.2% sodium azide (Carl Roth, Karlsruhe, Germany) and incubation took place at 4 °C for 24 h. The following day, the embryos were washed three times for 1 h in blocking solution containing 0.5% TritonX 100 (Ferak Berlin, Berlin, Germany) at RT. The embryos were then incubated with the HRP-labelled secondary anti-rabbit IgG antibody (Medac Diagnostics, Wedel, Germany), diluted 1:500 in blocking solution, at 4 °C o/n. Afterwards, they were washed three times for 1 h in blocking solution containing 0.5% TritonX 100 (Ferak Berlin, Berlin, Germany) at RT.

Embryos were incubated for 1 h in the DAB staining substrate solution (Vector Laboratories, Burlingham, CA, USA). Subsequently, 1 drop of H_2_O_2_ was added and incubated for another 10 min. The staining reaction was stopped by rinsing in PBS three times. The embryos were postfixed in a 2% PFA solution o/n at 4 °C. Finally, they were dehydrated in ascending alcohol series of PBS, 50% methanol, 80% methanol for 30 min each and stored in 100% methanol at -20 °C. Every incubation step in the protocol was performed under gentle shaking. The pictures of the embryos were taken with a digital camera.

### 2.10. Statement about Sample Size Estimate, Blinding and Statistics

The sample size is indicated in each figure caption. Embryo assay readouts were a matter of yes and no, and so, power analysis for sample size was not necessary. Blinding was ensured, as genotyping was performed after the analyses. No statistical methods were used.

## 3. Results and Discussion

### 3.1. Spatial Expression Pattern of MORG1 Protein in Organs of Adult and Embryonic Mice

A search in the human protein atlas (https://www.proteinatlas.org/ENSG00000123154-WDR83/tissue; accessed on 2 July 2022) revealed that MORG1 (WDR83) protein is expressed at varying levels in almost all human organs and tissue systems. For example, in the brain, glial cells of the basal ganglia, hippocampal formation and cerebral cortex express MORG1 weakly, while Purkinje cells of the cerebellum expressed it moderately strong. Similarly, lung cells, liver cells, cardiomyocytes and tubule cells of the kidney were reported to express MORG1 moderately, while cells of the pancreas showed weak expression. In this study, immunohistological staining for MORG1 confirmed, for the first time, a widespread expression also in murine tissues (Figure 1A). Expression of MORG1 can be observed in brain, lung, kidney, heart, liver and pancreas tissue of adult mice. Glial cells, cardiomyocytes, islet cells of pancreas and proximal tubule cells of the kidney showed especially intense staining, whereas murine liver cells express MORG1 weakly.

To investigate the spatial expression of MORG1 in mouse embryos, whole-mount staining as well as IHC staining of paraffin sections for MORG1 on E12.5 wild-type embryo was performed (Figure 1B–D). In the whole mount staining, an intense staining of the neural tissue in the brain ventricles and developing organs could be observed and the liquid-filled ventricles were clearly visible in the embryos. There was no expression of MORG1 observed in the limbs (Figure 1B). In stained tissue sections, MORG1-immunoreactivity was detected in all developing organs (Figure 1C). Very high level of MORG1 expression was found in the neuroepithelium of the neural tube as well as in the trigeminal and dorsal root ganglia (Figure 1D). In addition to neuronal structures, the otic cavity and tail were also strongly positive. Intense staining was also observed in extraembryonic tissues, including in trophoblast cells (Figure 1C,D). In contrast to shown IHC of adult organs, MORG1 expression appeared stronger in liver cells and weaker in heart cells (compare Figure 1C with Figure 1A).

To obtain further evidence, whether MORG1 may be associated with proliferation and differentiation programs, we co-stained for Ki67 or WT1 (Figure 2). Ki67 is widely used as a proliferation marker. Although it is now clear that Ki67 is not required for proliferation, it was, nevertheless, highly expressed in cells undergoing mitosis [15]. The Wilms’ tumor suppressor gene 1 (Wt1) was critically involved in a number of developmental processes in vertebrates, including cell differentiation, control of the epithelial/mesenchymal phenotype, proliferation and apoptosis [16]. As expected, the expressions of proliferation and differentiation markers Ki67 and WT1 did not overlap (Figure 2B–D). Furthermore, Ki67-positive cells were mainly found in the top layer of the neuroepithelium, endothelium and liver and WT1-positive cells mainly in the decidua, epicardium of the heart and the liver mesothelium (Figure 2B–D). In the literature, it was reported that Wt1 expression in the heart was predominantly, but not exclusively, associated with epicardial development. The earliest expression of Wt1 during cardiac morphogenesis was detected in mouse embryos at E9.5 in the proepicardium, which was the epicardial primordium; subsequently, Wt1 expression continues during the epicardial covering of the heart [16]. Wt1 was expressed in the liver mesothelium from the early stages of hepatic development. Liver mesothelial cells continued to express Wt1 when they migrated from the surface and intermingled with the hepatoblasts and the hematopoietic cells to differentiate into sinusoidal endothelium and stellate cells [16]. We observed partial colocalization of Ki67- and MORG1-positive cells (Figure 2E). Colocalization was particularly evident in the neuroepithelium and liver. Ki67-positive cells that did not express MORG1 were mainly found in the endothelia. In contrast to Ki67 staining, WT1-positive cells did not show any MORG1 co-expression (Figure 2F). These data supported a role of MORG1 in actively proliferating cells but not in cellular differentiation processes.

To explore MORG1 distribution earlier in development, we resorted to publicly available datasets. The EMBL-EBI database (https://www.ebi.ac.uk/gxa/genes/ensmusg00000005150; accessed on 5 June 2023) documented MORG1 expression as early as E8.0 in trophoblast cells. According to the eGastrulation database (egastrulation.sibcb.ac.cn), MORG1 expression is widespread across embryonic tissues between days E5.5 and E7.5, with a particularly high expression in the ectodermal layer (from which, among other things, the central nervous system develops) (Appendix A).

In summary, we concluded that MORG1 exhibits broad embryonic tissue distribution in mice, similar to data reported in humans, with a particular high expression in the neuronal system.

### 3.2. Generation of Morg1 Knockout Mice and Characterization of Offspring

MORG1 was identified in a wide variety of physiological signaling pathways and pathological processes and, therefore, represents an interesting target in biomedical research. Morg1 knockout mice were generated by homologous recombination [17]. Morg1 was disrupted by a construct shown in Figure 3A. Genomic DNA was obtained from tail biopsies and genotyping was performed by PCR to detect the mutated and the wild-type alleles (Figure 3B). Heterozygous (+/−) mice were fertile and had no apparent phenotypes when compared to the wild-type (+/+) littermates.

Long-term observation revealed that Morg1+/− mice showed normal development of both sexes, and the oldest mice survived beyond 25 months of age without evidence of significant spontaneous disease. There were also no differences between heterozygous and wild-type mice in the different age groups regarding body weight (own observations).

### 3.3. Homozygous Morg1 Knockout in Mice Leads to Embryonic Lethality

Morg1+/− were intercrossed to characterize phenotypes of Morg1 homozygous (−/−) knockout mice. We monitored several generations and found that approximately 65% of the offspring were Morg1+/− and 35% were Morg1+/+ (Table 1). Yet, we never detected live-born Morg1−/− offspring, indicating that homozygous deletion of Morg1 leads to embryonic lethality in mice. To further determine the developmental stage of embryonic death, the genotypes of embryos from embryonic days E8.5 to E12.5 from Morg1+/− intercrosses were analyzed (Table 1).

Genotyping embryos at very early stages is challenging because no material from the embryos should be lost before the studies and contamination with maternal tissue must be avoided. To address both reservations, different methods were used in this study. The examination of embryonic tissue from paraffin sections as well as Reichert’s membrane in early gestation revealed the presence of Morg1−/− embryos (Figure 3C,D). From E8.5 to E10.5, Morg1−/− embryos were seen at the expected frequency of a single-gene inheritance model (Mendelian distribution of 1:2:1). On E11.5 and E12.5, only embryos with the genotypes Morg1+/+ and Morg1+/− could be unambiguously identified, since all other embryos found were already dead in utero. On E11.5 20% and on E12.5 21% of the litters, resorptions could be identified. There was a lack of material to genotype the embryonic residues unambiguously due to the advanced stage of resorption. Regarding the altered distribution of the genotypes after E10.5, we assumed that these embryos were likely Morg1−/−. These results indicate that embryonic lethality of Morg1−/− progeny occurred already in the first half of pregnancy prior to E11.5 leading to resorption in the mid-gestational stage. This was not unusual, because although conventional knockout mice are suitable genetic models for inherited diseases, they often exhibit embryonic or early postnatal lethality [18]. Approximately 25–30% of complete gene knockouts in mice cause intra-uterine lethality [19].

Our macroscopic examination of implantation sites of Morg1+/− intercrosses identified an underdevelopment of Morg1−/− embryos from E9.5 onwards and advanced resorption from E11.5. On E8.5, no macroscopic differences between the genotypes could be seen. All the implantation sites showed a physiological appearance and did not differ in size, color or shape (own observations). Already on E9.5, macroscopic differences between the genotypes could be seen: the Morg1−/− embryos were clearly smaller (Figure 3E). On E10.5, the implantation sites occupied with Morg1−/− embryos were significantly smaller than their healthy littermates and featured hemorrhages, as identified by the dark coloring (Figure 3F). Until E11.5, some continued to shrink and blood seemed to leak and accumulate further in the implantation sites, and on E12.5, only small dark resorption sites remained as a sign of aborted pregnancy (own observations). Due to the high proportion of resorptions and the fact that no more knockout embryos could be detected after E10.5, we assumed that the resorbed sites contained Morg1−/− embryos.

### 3.4. Resorption of Embryonic Tissue on E12.5 and E11.5 as Well as Apoptosis and Decomposition of Embryos on E10.5 and E9.5 in the Morg1−/− Genotype

The first indication of prenatal lethality is a deviating Mendelian distribution between the genotypes of heterozygous intercrosses, accompanied by a smaller litter size compared to control groups. If there is evidence that no viable homozygous offspring are born, the investigation of key stages of gestation is recommended [20]. Examination of a litter on E12.5 provides a good initial point in time. If there are implantation sites or embryonic residues detectable, the offspring died after implantation on E4.5. From the progression of resorption, the approximate time of death can then be deduced.

At E12.5, healthy wild-type embryos were approximately 7–9 mm in size and their organs were already present and continue to differentiate into functional units. In contrast, the resorption of the potential Morg1−/− embryo and most of the surrounding tissue was already completed at this time-point (own observations). Only fragments of the extraembryonic tissue, such as the ruptured Reichert’s membrane, could be identified in the former embryonic cavity. The resorption was characterized by massive evasion and clotting of maternal blood. The hemorrhagic blood clot contained numerous immune cells and was interspersed with fibrous material. Only at the edge of the dissolved or liquefied placental tissue decidual tissue was detectable. The extensive areas of infiltrating immune cells were surrounded by fibrinoid material, cell debris and mucus.

About 21% of E12.5 and 20% of E11.5 embryos were in a state of advanced resorption. Due to the previous apoptosis, infiltration and decomposition of Morg1−/− embryos, it was assumed that they underwent degradation and subsequent resorption between E10.5 and E11.5. On E11.5, the physiologically developed embryos measured around 6–7 mm and all organs were already present in their primary structure. No such structures could be identified during the examination of the small and darkly colored Morg1−/− embryos on E11.5 (own observations). While the uterine epithelium, spongial and labyrinth layer were still visible, the embryonic tissue was already resorbed. However, the extraembryonic tissue displayed numerous apoptotic figures and underwent cytolysis. Extensive hemorrhages, consisting of erythrocytes and a high proportion of immune cells, were formed by bleeding from maternal vessels. Eosinophiles, which are involved in degradation of fibrous tissue, were particularly prominent in these areas. Massive accumulation of granulocytes populated the fading and disintegrating tissue and formed a purulent focus in the former embryonic cavity. Isolated shrunken apoptotic cells could be spotted between the necrotic tissue containing purulent debris.

In contrast, Morg1−/− embryos were still detectable in the uterine cavity at E10.5, albeit much smaller compared to the wild type Morg1+/+. However, infiltration of inflammatory cells and destruction of embryonic tissue was already evident (Figure 4A).

The mutant placenta took up less space but there was no evidence of cell death or excessive infiltration at this time (Figure 4A). The maternal moiety of the implantation site appeared intact. The embryos, on the other hand, were at a stage of late apoptosis, identified by nuclear fragmentation, and showed a loose and irregular cell association with barely differentiable tissue structures (Figure 4A). The embryonic cavity and embryo itself were extensively infiltrated by various immune cells, especially granulocytes, which was accompanied by tissue decomposition. The neuroepithelium of the degenerating embryo was irregular and showed large gaps in cell structure compared to the wild-type Morg1+/+ (Figure 4A). No organs were discernible due to impaired development and autolysis.

At E9.5, Morg1−/− embryos were significantly smaller than their healthy littermates and featured a defective turning into the final fetal position, already pointing to underdevelopment (Figure 4B). The formation of organ primordia had progressed poorly and the embryos were already in an early stage of apoptosis, displaying nuclear fragmentation and cellular gaps. The neuroepithelium especially showed extended intracellular spaces with sporadic bleb forming.

### 3.5. Growth Retardation and Maldevelopment of Morg1−/− Embryos on E9.5 and E8.5

Since the decomposition of the embryos was well advanced by day 10.5, we examined earlier stages to understand the underlying malformations by ex vivo MRI. In T2 weighted images, liquid water would usually be displayed very bright and other tissue in various shades of gray. However, the diffusion weighting due to the varFlipRARE sequence almost completely suppresses the signal from liquid water and creates a strong contrast to all other tissues. Using the medical image viewer OsiriX, the embryos could be rotated and viewed around all axes. Figure 4C shows representative images of Morg1+/+ and Morg1−/− embryos on E9.5. Morg1+/+ embryos turned in the final fetal position where the back was facing the membranes surrounding the embryo. Organ primordia and brain ventricles were visible in the sagittal plane. In the transversal orientation, a closed neural tube and the primordia of the limbs could be seen. Morg1−/− embryos appeared smaller and the tissue was unstructured and dissolved. In the sagittal plane, the embryonic vesicle was disrupted and embryonic tissue seemed to be moving out of the embryonic cavity (Figure 4C).

Stereomicroscopic images of extracted Morg1−/− embryos also showed malformation on E9.5 compared to the wild-types (Figure 4D). At E9.5, no brain ventricles were visible in Morg1−/− embryos, they exhibited a profound reduced tissue mass and lacked the physiologically curved embryonic position and failure of chorioallantoic fusion. Of note, there are several other examples of KO mice with failed chorioallantoic fusion in this embryonical development window of lethality [21,22]. In contrast to Morg1+/+ embryos, where several vessels and especially the aorta dorsalis could be seen immediately ventral to the somites, the formation of large vessels was poor or even absent in Morg1−/−, indicating impaired vascularization. Another abnormality observed on E9.5 was a very prominent layer of TGCs surrounding the Morg1−/− embryos, occupying significantly more placental space than in healthy offspring (Figure 4E).

Even at E8.5, Morg1−/− embryos showed growth inhibition and advanced underdevelopment (Figure 5). The MRI examination identified growth-inhibited embryos that were later confirmed as Morg1−/−. The average embryonic cavity size of Morg1−/− mice was about 76% reduced compared to the wild-type and heterozygous animals at E8.5. Morg1−/− embryos were also extremely small compared to average wild-type embryos and at least one day behind the physiological developmental stage. With this MRI resolution, no specific structures but just a relatively thin tissue layer was visible in the uterine cavity of Morg1−/− embryos, whereas in the Morg1+/+, a dorsal bending of the neuroepithelium to form the neural folds could be clearly observed (own observations). Embryos, which were first analyzed by MRI, were fixed, embedded in paraffin and subjected to IHC analysis. H&E staining confirmed the developmental delay in mutants (Figure 5A). Morg1+/+ embryos exhibited around eight pairs of somites, an optic vesicle, a thick and uniform neuroepithelium and development of the heart, with cardiac looping of the linear heart tube. Morg1−/− embryos lacked most of these developmental features. Their neuroepithelium was still thin and irregular. The head region was underdeveloped with no discernible head fold and neither a cardiac primordium nor somites were recognizable in the section. Furthermore, mutants had a clearly smaller cell mass and cell density and their embryonic tissue showed anomalous organization and showed failure of chorioallantoic fusion.

### 3.6. Severe Placental Abnormalities and Delayed Neural Tube Closure in Morg1−/− Embryos

Early embryonic lethality between E9.5 and E14.5 is commonly associated with placental malformations that lead to undersupply and cause maldevelopment. In a study of 103 embryonic lethal and subviable (strains in which the proportion of mutant offspring is >0% but ≤13%) knockout lines, 68% of the lines that were lethal at or after mid-gestation showed placental dysmorphologies [19]. Proper placental vascularization is essential for the physiological development of the embryo and surrounding tissues [23]. Embryonic malformations showing significant statistical correlation with placental defects included abnormalities in the heart, brain and vascular system. Effects of placental insufficiency on brain development manifested especially in the forebrain [19]. The embryonic trophoblast is essential for implantation and interacts with the maternal uterus. Trophoblast giant cells (TGC) initially make contact with the uterine cell lining and invade the uterine tissue [24]. Therefore, the invasion of vessels around the chorionic plate and labyrinth layer and the development of the TGC layer were investigated (Figure 4E and Figure 5B). In Morg1−/− embryos on E8.5, a severe impairment of placental vascularization was observed. In the wild-type placenta, the maturing labyrinth layer already contained numerous vascular beds full of erythrocytes, which were lacking in the Morg1−/− (Figure 5B).

To demonstrate an impact of Morg1 depletion on the developmental stage of the embryo, the tissue-specific expression pattern of key markers that both influence embryonic development and are associated with MORG1 manifestations was examined (Figure 5C). E8.5 Morg1−/− and Morg1+/+ embryos of the same litter were cut in sequence and the plane at which the neural tube was transversely sectioned was chosen for evaluation, because the development from the neural plate to the closed neural tube takes a crucial step at this time and serves as an indicator of physiological growth (Figure 5C, area marked with rectangle in top H&E images). The size of the uterus itself was not different, but the embryonic cavity of Morg1−/− embryos was significantly smaller. The thickness of the mutant neuroepithelium was reduced by about half and the neural groove was less advanced than in the Morg1+/+, indicating developmental delay. Failure of neural tube closure was uniformly observed in several Morg1−/− embryos. The process of neural tube morphogenesis is highly complex and apicobasal polarity mechanisms play a major role in neural tube closure [25]. The study of Hayase et al. showed that MORG1 is involved in the formation of cell polarization in epithelial cells. The formation of apicobasal polarity in epithelial cells involves atypical protein kinase C (aPKC), which constitutively interacts with Par6, an evolutionarily conserved adaptor protein. [8]. The Par6-aPKC complex is shown to translocate from the cytoplasm to the apical membrane, where the complex may be anchored with the apical membrane-integrated protein Crumbs3 (Crb3), which is reinforced with the apically localized small GTPase Cdc42 in the GTP-bound form [8]. For many years, the mechanism for Par6-aPKC translocation to the apical membrane was unclear, but then, it was shown that the forced targeting of the complex to the apical surface is mediated by MORG1 [8]. This conclusion is based on the findings that MORG1 directly binds not only to Par6 but also to Crb3, which facilitates Par6 binding to Crb3, leading to apical targeting of Par6-aPKC [8].

To ensure rapid growth during this phase of embryogenesis, a high proliferation rate in embryonic tissue is indispensable. In the wild-type embryos, a high expression level of Ki67 was detected in all embryonic tissues and especially in the neural tube (Figure 5C), which is typical around E8 [26]. In contrast to E12.5, where Ki67-positive cells were now restricted to only a thin layer of neuroepithelium (Figure 2B), they were distributed throughout the neuroepithelium at day E8.5 (Figure 5C). MORG1 expression was also uniformly distributed in the neuroepithelium in wild-type at E8.5, which did not change at days E9.5 and E10.5 (own observations). In Morg1−/− embryos at E8.5, the proliferation rate decreased considerably and only very few cells were positive for Ki67, indicating stagnant proliferation and development. Thus, it is reasonable to assume that cell cycle arrest occurred in most cells of the embryonic tissue at that stage and that further development and progression came to a standstill.

Because a known role of MORG1 is to stabilize PHD3 and increase oxygen-dependent degradation of HIF, we examined the expressions of PHD3 and HIF-1α and -2α. Although PHD3 was not affected by knockout of Morg1, lower levels of both HIFα isoforms were present in Morg1−/− embryos and extraembryonic tissues (Figure 5C). Former nephrological studies with adult mice did show that in heterozygous Morg1 mutants, a loss of PHD3 stabilization leads to reduced PHD3 concentration and, therefore, reduced HIFα degradation [7], but this was not comparable to the present study. Apart from the fact that this study examined adult kidney tissue from Morg1+/− mice and early embryonic stages in homozygous knockout, there are other PHDs besides PHD3, namely PHD1 and 2, that regulate the stability of HIFα subunits. In any case, during embryonic development, other biological functions of MORG1 seem to play a role rather than that of being a scaffolding protein of PHD3. An influence of MAPK signaling pathway on HIF expression came into focus as a possible factor, because a requirement of the ERK-pathway for HIF-1 induction was described in neurons and a direct phosphorylation of HIF-1 by ERK 1/2, leading to an increased half-life, was reported [27,28]. Since MAP kinases are essential for neuronal migration during brain development [29,30], a disruption of proper MAP kinase function by deletion of the scaffolding protein MORG1 may easily explain the severe neuronal development defects. For example, disruption of the ERK2 gene leads to impaired trophoblast development and post-implantation lethality [31] and knockout of MEK1 causes embryonic death on E10.5 due to impaired vascularization in the labyrinth layer of the placenta [32]. The observed phenotypic changes of Morg1 knockout offspring appear to manifest similarly to suppressions of downstream signaling molecules of the MAPK pathway.

Coming back to the role of HIF, before the onset of placental circulation around E8/E9, the embryos reside in a hypoxic environment and both HIF-1α/2α are expressed at high levels [33]. Beyond this time point, the expression of HIF-1α and HIF-2α vary widely depending on tissue and time point of gestation. However, the exact reason for the divergent expression patterns remains unclear to date [34,35]. The reported phenotypes of HIF-1α and/or HIF-2α knockout embryos fit largely with the observations of the Morg1−/− embryos reported here. HIF knockout embryos showed angiogenic and neural tube defects and cardiovascular malformations, and they underwent developmental arrest by E9 and lethality by E10.5 [36]. Furthermore, the placenta of a double knockout of HIF-1α and -2α exhibited no fetal blood vessels, expanded TGC numbers and aberrant placental architecture [36], proving the importance of HIF on proliferation, survival and differentiation of multiple embryonic tissues.

VEGF-A is essential for embryonic and placental development due to its major role during hematopoiesis, vascularization and angiogenesis [37]. During murine embryogenesis, it can be detected from E7 in the extraembryonic and embryonic ectoderm. By E8.5, VEGF-A is present especially in the trophoblast surrounding the embryo, the embryonic myocardium, gut endoderm, embryonic mesenchyme and amniotic ectoderm and, later, in the neuroectoderm of the head [38]. In accordance with these literature data, on E8.5, VEGF-A was highly expressed in the Morg1+/+ trophoblast surrounding the embryo and the embryo itself (Figure 5C). The Morg1−/− embryos lacked VEGF-A expression and the staining of the trophoblast cells was less intense than in the wild-type. Previous studies indicated that HIF-1α is involved in mediating VEGF transcriptional activation in hypoxic cells [34].

In conclusion, the reduced expression of VEGF-A, HIF-1α and HIF-2α symptomatically matched the impaired vascularization of the placenta and very likely played a role in the progression of embryonic malformation. Proliferative arrest preceded subsequent apoptosis and decomposition of the embryo. Since hardly any proliferating cells were present on E8.5, it can be assumed that undersupply was previously relevant in underdevelopment.

CD31, widely used as an endothelial cell marker, indicates the presence of vascularization and angiogenesis [39]. Both genotypes showed the same expression of CD31 in intensity and localization (Figure 5C). Expression appeared in the embryo itself and all surrounding tissues including trophoblastic layer and decidua. Thus, we concluded that impaired expression of CD31 was not a likely cause for the insufficient formation of vessels in Morg1−/− embryos.

Embryonic and extraembryonic tissues undergo constant remodeling, which is not possible without expression of proliferative and apoptotic markers [40]. Cleaved caspase-3 represents a reliable marker for early detection of apoptotic cells because it is responsible for most of the proteolysis during apoptotic processes and activated during the intrinsic and extrinsic apoptosis signaling pathway [41]. Immunofluorescence antibody staining for cleaved caspase-3 of the embryo itself and the layer of TGCs surrounding the embryonic vesicle are shown in Figure 5D. In the Morg1+/+ embryos on E8.5, no cleaved caspase-3 positive cell could be identified. When examining the Morg1−/− embryos, approximately one caspase-expressing cell was found per section, showing no unphysiological signs. However, strong activation of caspase-3 was observed in the Morg1−/− trophoblastic layer. Even if there were no signs of cell death in the embryos, there already seemed to be strong deviations from the Morg1+/+ in the TGC. Advanced apoptosis from E9.5 could be identified in the H&E staining (Figure 4B), as described previously.

Taking all findings together, we document here that the intrauterine death of Morg1−/− embryos is most likely caused by a severe failure to develop brain and other neuronal structures such as the spinal cord and ganglia. The pivotal event of the central nervous system, namely the formation and closure of the anterior neuropore, normally occurs at day E9.0 [42]. We previously found that MORG1 is expressed in the normal human brain and is downregulated following ischemic brain damage [43]. However, MORG1 expression was high in reactive astrocytes adjacent to the ischemic brain areas [43]. Furthermore, in an experimental model of brain ischemia (MCAO), a comparison of Morg1+/− mice to the wild-type revealed a significantly reduced infarct volume despite a similar restriction of blood flow in both mice phenotypes as measured by laser Doppler flowmetry [44]. These findings clearly point to a central role of MORG1 in astrocyte and/or neuronal development and remodeling.

## 4. Conclusions

In summary, Morg1−/− embryos on E8.5 and E9.5 showed failure to turn, defective chorioallantoic fusion, growth inhibition, maldevelopment, very low cell mass, poorly differentiated tissue, impaired vascularization and heart development and a particularly thin and irregular neuroepithelium of the underdeveloped neural tube. Given the regulatory role of MORG1 on the HIF and MAPK pathways as well as of apicobasal polarity mechanisms, and since these were shown to play a critical role in embryonic development, disruption of these pathways is likely to be causally involved. Maldevelopment of the surrounding extraembryonic tissue may further lead to an undersupply of the Morg1−/− embryo, which is reflected by proliferative arrest on E8.5. The presented results underpin a major role of MORG1 in nervous system development. Thus, and given the multifaceted molecular biological influence and wide expression of MORG1, it is reasonable to speak of a multifactorial process that causes embryonic death in mutant mice. More extensive studies, especially to changes in ERK phosphorylation/nuclear translocation and the polarization of neuroepithelial cells in the homozygous Morg1 knockout, are needed to further analyze the influence of MORG1 during mammalian embryonic development and its crosstalk with placental tissue.

## Figures and Tables

**Figure 1 biomolecules-13-01037-f001:**
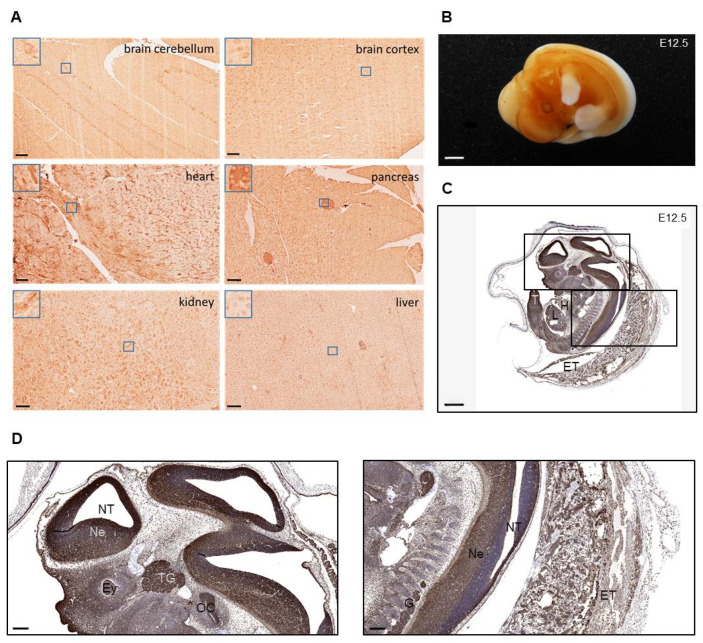
Immunohistochemical MORG1 staining of Morg1+/+ adult tissue and embryos. E = embryonic day; ET = extraembryonic tissue, Ey = eye; H = heart; L = liver; Ne = neuroepithelium; NT = neural tube; OC = otic cavity; T = tail; TG = trigeminal ganglion. (**A**) Staining of adult wild-type murine tissues showing the distribution of MORG1. MORG1 positive cells in brain cerebellum and cortex, in cardiomyocytes in heart, islets in pancreas, hepatocytes and proximal tubules in kidney. Scale bar = 100 µm (**B**) MORG1 whole mount staining of Morg1+/+ embryo on E12.5. N = 2. Scale bar = 800 µm (**C***,***D**) Simulated DAB/Hematoxylin image (Pathology Views^TM^) from immunofluorescence of MORG1 on E12.5. (**C**) Particularly high expression in neuroepithelium of the neural tube and brain vesicles, ganglia and organ primordia. Scale bar = 800 µm (**D**) Left picture displaying embryonic neural tissue with high expression of MORG1 in neuroepithelium, the trigeminal ganglion and otic cavity. Right picture showing neural tube and extraembryonic tissues with high expression in ganglia. Scale bar = 200 µm.

**Figure 2 biomolecules-13-01037-f002:**
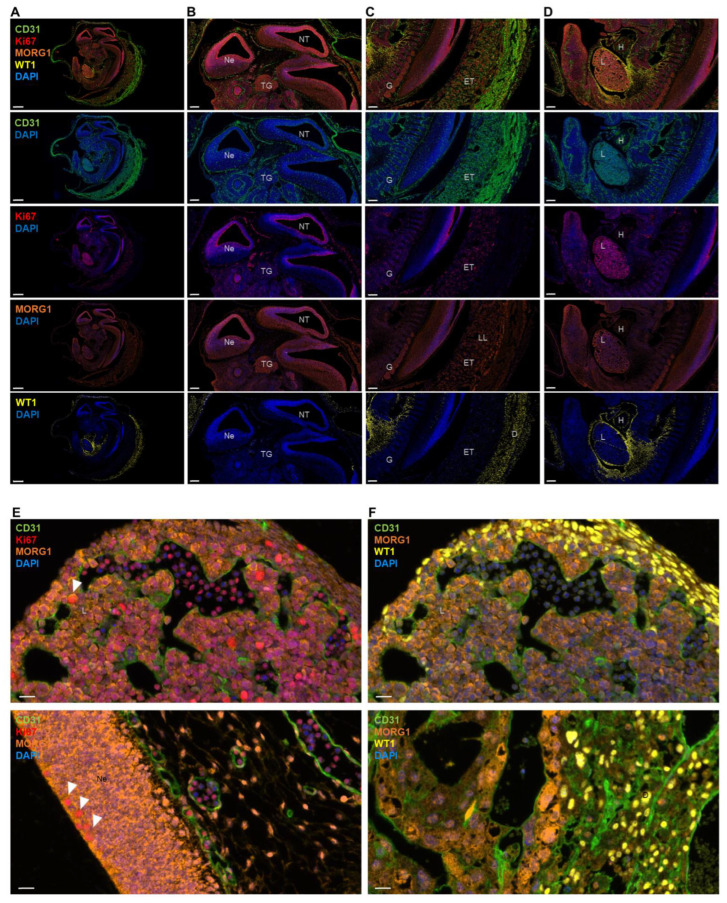
4-plex immunofluorescence with CD31, Ki67, MORG1 and WT1 of Morg1+/+ E12.5 mouse embryo. D = decidua; E = embryonic day; ET = extraembryonic tissue, H = heart; L = liver; LL = labyrinth layer; Ne = neuroepithelium; NT = neural tube; TG = trigeminal ganglion. (**A**) Image of the whole embryo with combined and single antibody staining shown. Scale bar = 800 µm (**B**) Image of the embryonic neural tissue. The vascular marker CD31 is highly expressed in vessel endothelium. Ki67 is detected in dividing cells in the top layer of the neuroepithelium of the embryo. MORG1 is expressed in many tissues like the neuroepithelium and the trigeminal ganglion. No expression of WT1 in the neuroectoderm was found. Scale bar = 200 µm (**C**) Tissues surrounding the embryo shown. CD31 was detected in the highly vascularized extraembryonic tissue, while there was only few mitoses shown with Ki67. MORG1 is expressed extensively in the ganglia and the labyrinth layer of the placenta. Scale bar = 200 µm (**D**) Throughout the mid-section of the embryo, particularly in the developing liver, CD31 was detected where Ki67 is also most abundant. A stronger expression of MORG1 could be found in the liver than in the heart of the embryo. Contrasting MORG1, WT1 is mainly expressed in the mesothelial epithelium lining coelomic cavities. It can be found in the epicardium of the heart and the liver mesothelium. Scale bar = 200 µm (**E**) Ki67 seems to partially colocalize with MORG1 (white arrows). Colocalization was particularly evident in the liver (upper panel) and neuroepithelium (lower panel). Scale bars = 20 µm (**F**) WT1 shows no colocalization with MORG1, neither in the liver (upper panel) nor in the decidua (lower panel). Scale bars = 20 µm.

**Figure 3 biomolecules-13-01037-f003:**
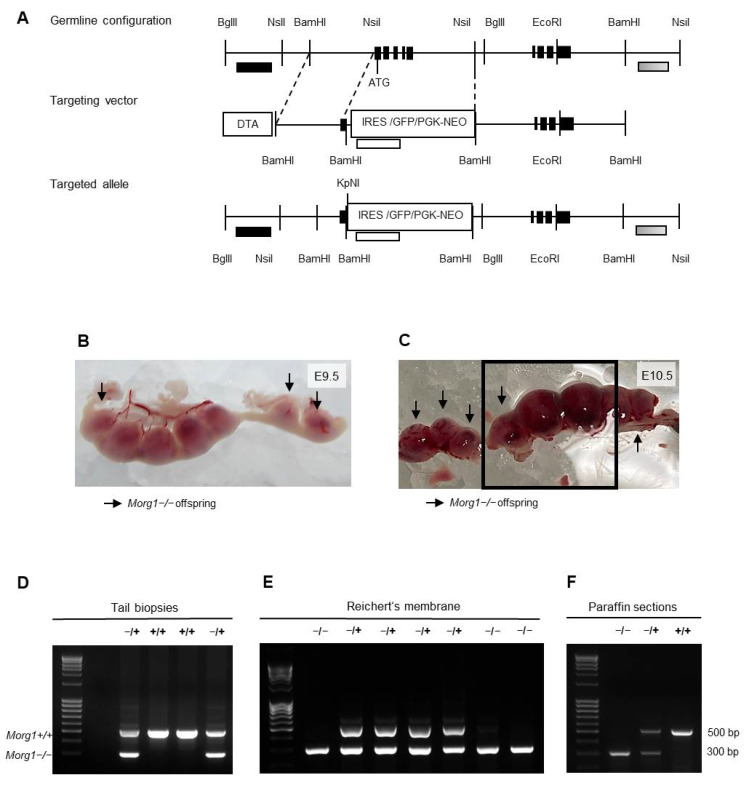
Generation of Morg1−/+ mice and Morg1−/− embryos and genotyping of offspring. bp = base pair; DTA = diphtheria toxin; E = embryonic day; GFP = green fluorescence protein; IRES = internal ribosomal entry site; NEO = neomycin phosphotransferase; PGK = phosphoglycerate kinase. (**A**) Structure of Morg1 knockout construct. Schematic illustration of germline configuration, targeting vector and targeted allele shown. (**B**) Genotyping results of adult mice using material obtained from tail biopsies. Only wild-type (Morg1+/+) and heterozygous (Morg1−/+) mice were born alive. (**C**,**D**) Genotyping results of embryos using material obtained from Reichert’s membrane (**C**) or paraffin sections (**D**). Homozygous knockout (Morg1−/−) embryos were identified in addition to wild-type (Morg1+/+) and heterozygous (Morg1−/+) genotypes. (**E**,**F**) Uteri extracted on E9.5 and E10.5 from Morg1−/+ intercrosses. Implantation sites containing Morg1−/− embryos (arrow) appear smaller and hemorrhagic on E10.5. Of note, the genotypings in C and D correspond to embryos from E and F in the same order.

**Figure 4 biomolecules-13-01037-f004:**
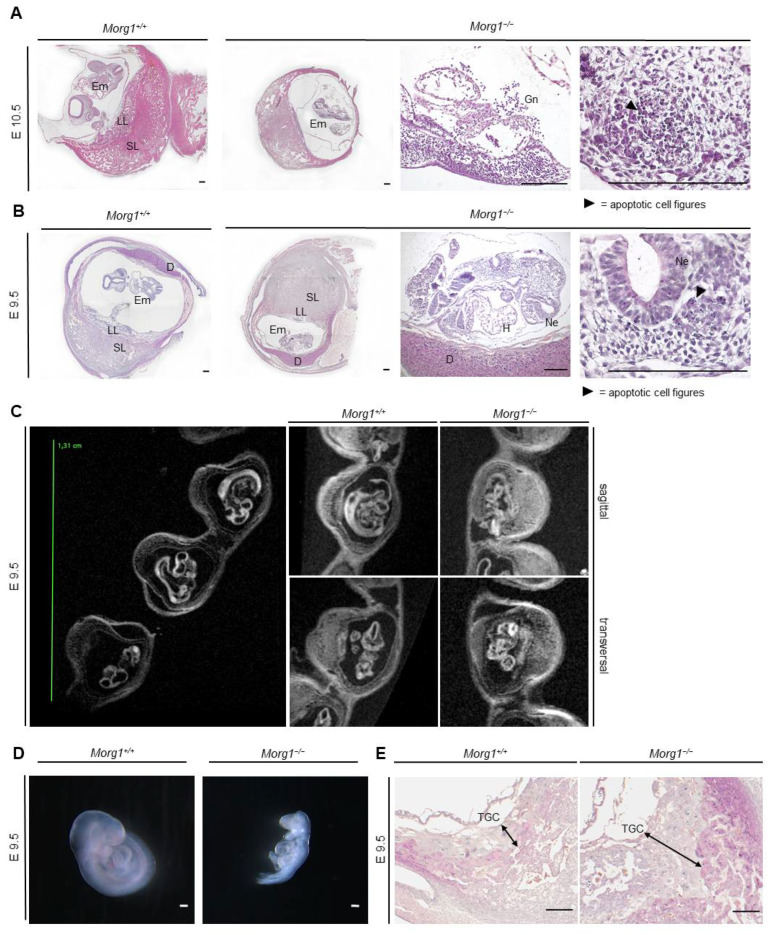
Apoptosis and decomposition of Morg1−/− embryos. H&E staining. D = decidua; E = embryonic day; Em = embryo; Gn = granulocytes; H = heart; LL = labyrinth layer; Ne = neuroepithelium; SL = spongial layer; TGC = trophoblast giant cell. Scale bar = 200 µm. (**A**) Decomposition of Morg1−/− embryo on E10.5 compared to healthy Morg1+/+ embryo. Morg1−/− embryo displays apoptosis, loose cellular structure and massive infiltration of immune cells. The maternal moiety of the implantation site seems intact. N = 3 each of wild-type and knockout. (**B**) Underdeveloped Morg1−/− embryo is smaller than the healthy Morg1+/+ embryo and exhibits defective turning into the final fetal position and failure of chorioallantoic fusion. The Morg1−/− embryo is in an early stage of apoptosis, displaying nuclear fragmentation, loose cell cluster and cellular gaps, especially in the neuroepithelium. N = 3 each of wild-type and knockout. (**C**) Representative MRI images of Morg1+/+ and Morg1−/− embryos together as implantation sites in the uteri (large image, on the left) or single in sagittal and transversal layer (small image, on the right). Morg1−/− embryo exhibits growth retardation, advanced malformations and unstructured tissue appearance on E9.5. N = 9 (of which 2 knockout). (**D**) Stereomicroscopic images of unprocessed Morg1+/+ and Morg1−/− embryos of the same litter. Morg1−/− embryos are underdeveloped, much smaller and lack developmental features. The embryos failed to turn in curved embryonic position. N = 10 (of which 3 knockout). (**E**) H&E staining of extraembryonic tissue. Morg1−/− placenta exhibits extended layer of TGCs on E9.5 and shows impaired vascularization including diminished formation of vascular beds. N = 3.

**Figure 5 biomolecules-13-01037-f005:**
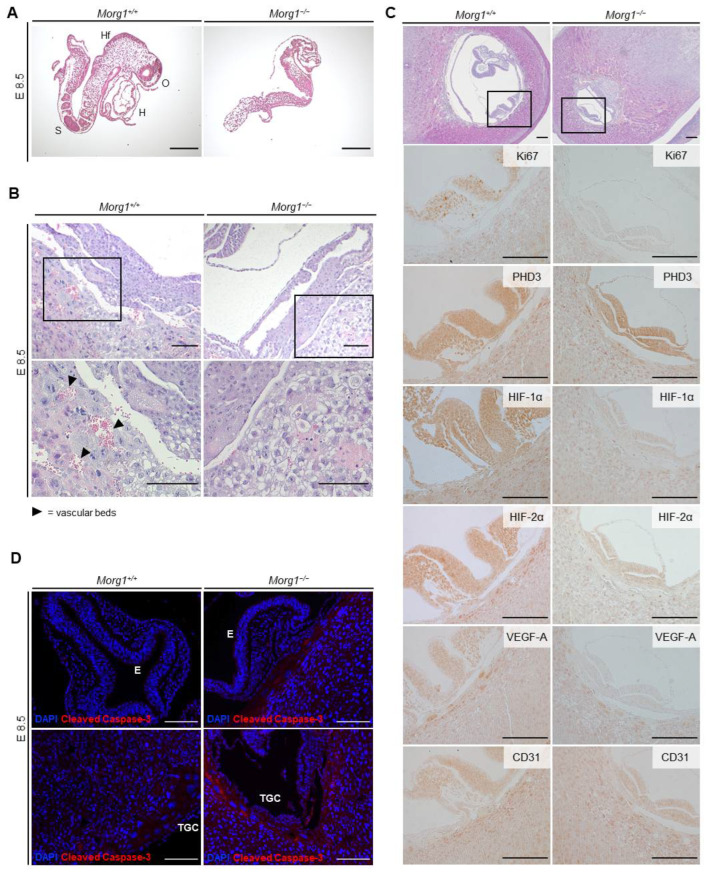
Maldevelopment of Morg1 knockout embryo on E8.5 and impaired vascularization through aberrant levels of HIF-1α, HIF-2α, VEGF-A, Ki67 and cleaved caspase 3 on E8.5. E = embryonic day; H = heart; HF = head fold; MRI = magnetic resonance imaging; O = optic vesicle; S = somites; TGC = trophoblast giant cell. Scale bar = 200 µm. (**A**) H&E staining of Morg1+/+ and Morg1−/− embryos on E8.5 of the same litter after preparation. The underdeveloped Morg1−/− embryos lack organ primordia and shows failure in chorioallantoic fusion. (**B**) H&E staining of extraembryonic tissue. Morg1−/− placenta exhibits extended layer of TGCs and shows impaired vascularization including diminished formation of vascular beds. N = 6. (**C**) H&E and IHC staining of Morg1+/+ and Morg1−/− embryo and extraembryonic tissue. Morg1−/− embryo exhibits impaired formation and development of neural tube and diminished expression of HIF-1α, HIF-2α, VEGF-A and Ki67. N = 5 each of wild-type and knockout with qualitatively similar stainings. (**D**) Representative immunofluorescence staining of Morg1+/+ and Morg1−/− embryo. TGCs surrounding the Morg1−/− embryo show elevated levels of activated caspase-3. N = 5 each of wild-type and knockout with qualitatively similar staining.

**Table 1 biomolecules-13-01037-t001:** Genotype analysis of offspring from Morg1+/− intercrosses. Numbers indicate observed embryos or born mice (in %) of each genotype at different stages of gestation. Expected percentage, based on Mendelian ratio 1:2:1, are in parentheses.

Embryonic Day	Total	*Morg1* Genotype			
		+/+	+/−	−/−	Unknown
E8.5	29	27.6% (25%)	44.9% (50%)	24.1% (25%)	3.4% *
E9.5	67	20.9% (25%)	46.3% (50%)	31.3% (25%)	1.5% *
E10.5	155	31.6% (25%)	46.5% (50%)	21.9% (25%)	0%
E11.5	80	26.3% (25%)	53.7% (50%)	0% (25%)	20% *
E12.5	38	23.7% (25%)	55.3% (50%)	0% (25%)	21% *
postnatal	75	34.7% (25%)	65.3% (50%)	0% (25%)	0%

* Unambiguous: genotyping was not possible in these embryos due to lack of material.

## Data Availability

Not applicable.

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
