# Peer review of "Targeted Disruption of the MORG1 Gene in Mice Causes Embryonic Resorption in Early Phase of Development"

_biomolecules, 2023, doi:10.3390/biom13071037_

Round 1

Reviewer 1 Report

This manuscript describes a careful and thorough analysis of the most recent MORG1 knockout mouse, this time focussing on the homozygous phenotype. Figure 5 presents the most insights with interesting changes in the expression of some putative target genes and pathways.

Experiments are well and thoroughly performed and the manuscript is well written. 

The manuscript would benefit a lot if the authors could include an analysis of a few apico-basal polarity markers at Day 8.5. The delayed neural tube closure seen is likely to be simply the result of a general developmental delay in the mutants but as MORG1 is also involved in cell polarisation of epithelial cells, the sections of the neuroepithelium might as well be investigated using some Par6 antibodies to examine if this pathway is defective and contributes to the defects seen particularly in this tissue. 

Author Response

This manuscript describes a careful and thorough analysis of the most recent MORG1 knockout mouse, this time focussing on the homozygous phenotype. Figure 5 presents the most insights with interesting changes in the expression of some putative target genes and pathways.

Experiments are well and thoroughly performed and the manuscript is well written.

We thank this reviewer very much for the positive evaluation.

The manuscript would benefit a lot if the authors could include an analysis of a few apico-basal polarity markers at Day 8.5. The delayed neural tube closure seen is likely to be simply the result of a general developmental delay in the mutants but as MORG1 is also involved in cell polarisation of epithelial cells, the sections of the neuroepithelium might as well be investigated using some Par6 antibodies to examine if this pathway is defective and contributes to the defects seen particularly in this tissue.

This is indeed a very interesting question and should definitely be addressed in future studies. We addressed this point in the manuscript (chapter 3.6, 2nd paragraph) and included a corresponding sentence in the conclusion. Unfortunately, we are not able to perform the necessary experiments to answer this question within the scope of this manuscript and the time available to us of 10 days until resubmission. We would need to obtain new embryonic material and procure appropriate antibodies and re-establish them for IHC.

Reviewer 2 Report

In this manuscript, Wulf et al examine the effect of MORG1 disruption during early mouse embryogenesis. This study of potential significance considering MORG1 is an important component of the MAPK/ERK signaling pathway and it's haplo-insufficiency results in several interesting phenotypes. However, it is unknown as to what happens upon complete loss of MORG1 during embryonic development. Overall, this paper describes the MORG1 mutant phenotype in great detail. I am supportive of this paper's publication provided the authors can address my following concerns: 

1) It appears that MORG1 function is required much earlier during development (prior to E8.5 since a phenotype is already evident by that stage in mutant embryos) and MORG1 appears to be required in the extra-embryonic region. To this end, it would be important to identify where MORG1 is expressed prior to E8.5. Is MORG1 expressed in the extra-embryonic region/trophoblast cells at E7.5? This can be shown either by IHC or by looking for MORG1 transcripts in already available scRNA-seq datasets of embryos at this stage. 

2) Specificity of the MORG1 antibody should be tested for and confirmed by using MORG1-/- embryos as negative control.

3) In Fig.2, higher resolution images showing co-localization of Ki67 and MORG1 (and panels showing WT1 and MORG1 in the same image) should be included to support the authors' claim that Ki67 and MORG1 are co-localized in certain cells and conversely in the case of WT1. 

4) Since MORG1 appears to be an essential component of MAPK/ERK signaling, it would be interesting to observe if there are any changes in this signaling pathway such as changes in ERK phosphorylation/nuclear translocation. 

Author Response

In this manuscript, Wulf et al examine the effect of MORG1 disruption during early mouse embryogenesis. This study of potential significance considering MORG1 is an important component of the MAPK/ERK signaling pathway and it's haplo-insufficiency results in several interesting phenotypes. However, it is unknown as to what happens upon complete loss of MORG1 during embryonic development. Overall, this paper describes the MORG1 mutant phenotype in great detail.

We thank this reviewer very much for the positive evaluation.

I am supportive of this paper's publication provided the authors can address my following concerns:

1) It appears that MORG1 function is required much earlier during development (prior to E8.5 since a phenotype is already evident by that stage in mutant embryos) and MORG1 appears to be required in the extra-embryonic region. To this end, it would be important to identify where MORG1 is expressed prior to E8.5. Is MORG1 expressed in the extra-embryonic region/trophoblast cells at E7.5? This can be shown either by IHC or by looking for MORG1 transcripts in already available scRNA-seq datasets of embryos at this stage.

We thank the reviewer very much for the extremely valuable hint to search the databases for MORG1 expression in earlier time points. In the EMBL-EBI database, we found information on trophoblast cells that were only from E8.0 embryos, but still detected Morg1 expression in this cell type. Interestingly, using the eGastrulation database, we found temporal-spatial expression of MORG1 in embryos from days E5.5-E7.5, showing that MORG1 is widely expressed across all zones at each stage. We have provided a new figure (suppl. Figure 2) for this purpose. Particularly impressive is the fact that MORG1 is already weakly to moderately expressed in the early embryos in the endoderm and mesoderm, but strongly to very strongly expressed in the ectoderm (from which, among other things, the central nervous system develops). This provides excellent support for our findings of the later time points.

2) Specificity of the MORG1 antibody should be tested for and confirmed by using MORG1-/- embryos as negative control.

Thank you for highlighting this point. We performed several controls to check the specificity of the MORG1 antibody. 1) the "negative tissue" control: for this we stained MORG1 knockout embryos. 2.) the "secondary antibody only" control we performed on both embryos and adult tissues. 3.) we performed the "isotype" control (IgG) in both embryos and adult tissues. 4.) The autofluorescence control of the mIFs results from the different and very specific individual staining. We have added a suppl. figure and method to the manuscript for this different control stainings. 

3) In Fig.2, higher resolution images showing co-localization of Ki67 and MORG1 (and panels showing WT1 and MORG1 in the same image) should be included to support the authors' claim that Ki67 and MORG1 are co-localized in certain cells and conversely in the case of WT1.

We thank the reviewer for pointing this out. We have inserted enlarged sections of co-stainings of MORG1 with Ki67 or with WT1 in Fig. 2E and 2F.

4) Since MORG1 appears to be an essential component of MAPK/ERK signaling, it would be interesting to observe if there are any changes in this signaling pathway such as changes in ERK phosphorylation/nuclear translocation.

We thank the reviewer for this comment. This is indeed a very interesting question and should definitely be addressed in future studies. We addressed this point in the manuscript (chapter 3.6, 4th paragraph) and included a corresponding sentence in the conclusion. Unfortunately, we are not able to perform the necessary experiments to answer this question within the scope of this manuscript and the time available to us of 10 days until resubmission. We would need to obtain new embryonic material and procure appropriate antibodies and re-establish them for IHC.

Round 2

Reviewer 2 Report

I thank the authors for their responses to my comments and the revised version of this manuscript  can be accepted.